# Functional Properties and Storage Stability of Astaxanthin-Loaded Polysaccharide/Gelatin Blend Films—A Comparative Study

**DOI:** 10.3390/polym14194001

**Published:** 2022-09-24

**Authors:** Katarzyna Łupina, Dariusz Kowalczyk, Waldemar Kazimierczak

**Affiliations:** 1Department of Biochemistry and Food Chemistry, Faculty of Food Sciences and Biotechnology, University of Life Sciences in Lublin, Skromna 8, 20-704 Lublin, Poland; 2Department of Biomedicine and Environmental Research, Faculty of Natural Sciences and Health, The John Paul II Catholic University of Lublin, Konstantynów 1J, 20-708 Lublin, Poland

**Keywords:** edible films, polysaccharides, gelatin, astaxanthin, colour stability, antioxidant activity

## Abstract

Edible films were obtained from the aqueous binary 75/25 blends of polysaccharides (carboxymethyl cellulose (CMC), gum Arabic (GAR), octenyl succinic anhydride starch (OSA), and water-soluble soy polysaccharides (WSSP)) and gelatin (GEL) supplemented with increasing concentrations (0, 0.25, 0.5, and 1% *w*/*w*) of water-soluble AstaSana (AST) astaxanthin. The AST-loaded films were red and exhibited a grainy microstructure and reduced transparency. The CMC- and WSSP-based films were the best UV-C blockers. After the incorporation of 1% AST, the antiradical activity of the films increased by 1.5 times (~25 percentage points) compared to the controls. The tensile strength (TS) of the CMC-containing films was much higher than those of the other films (36.88–43.04 vs. 2.69–15.62 MPa). AST decreased the TS of the CMC/GEL film (by ~11–14%) but improved the mechanical cohesiveness of the GAR/GEL film (by ~50%). The storage test (at 25 °C and 60 °C, no light access) revealed that the CMC- and GAR-based films exhibited the lowest colour change. Furthermore, at the elevated temperature, the films with higher AST concentration exhibited a better ability to maintain their colour. The WSSP/GEL films were the most prone to darkening and yellowing, possibly due to the Maillard reaction. Moreover, these films had the weakest antiradical activity.

## 1. Introduction

Active packaging refers to the incorporation of certain additives into packaging to maintain and extend product shelf life. As there are many oxidation-susceptible components in food, packages with an antioxidant effect are an important group of active packaging. In response to the growing problem with the disposal of used plastic packaging, many formulations of active packaging materials obtained from renewable raw materials, e.g., polysaccharides and proteins, have been developed. Depending on the method used, such packaging is biodegradable and edible, which creates a new application potential (e.g., edible coatings, casings, capsules, and microcapsules). Antioxidants introduced into food and edible packaging should primarily be highly effective (i.e., active at low concentrations), acceptable to consumers, and cheap. The current antioxidant market is very rich and offers newer and stronger substances. One of the most effective antioxidants is astaxanthin (ASX), i.e., a red carotenoid extracted from *Haematococcus pluvialis* algae or obtained with the use of synthetic methods due to the high demand. Studies have shown that natural astaxanthin is 54 times more potent than β-carotene, 65 times more potent than vitamin C, and 500 times more active than vitamin E and coenzyme Q10 [1]. With its antioxidant properties, ASX exerts a positive effect on human health [2,3,4]. Its values are appreciated by an increasing number of industries, including the research and development sector of the packaging industry [5,6,7]. It can be assumed that, owing to its red colour and antioxidant properties, ASX-supplemented packaging may be suitable for specific applications, e.g., red coatings for cheese, nuts, pork, beef, and surimi sticks [8].

As ASX is commercially available as a hydrophobic (e.g., oleoresin) as well as a hydrophilic (e.g., ASX derivatives) compound, the dual hydrophobic–hydrophilic behaviour of the carrier material could be a rational approach to the development of active packaging systems. Consequently, in this study, the polymers with emulsion-stabilizing effects were used as the ASX carriers. Due to certain advantages, AstaSana (AST)—a commercial formulation of water-soluble ASX—was used as the active ingredient. 

In materials science, a polymer blend, or a polymer mixture, is a member of a class of materials with extended useful properties beyond the range that can be obtained from a single polymer. Previous studies [9,10] have shown that a combination of polysaccharides and gelatin (GEL) yields materials with modified properties, including changed profiles of active compound release, which is extremely important for the efficiency of the active packaging. Our recent study showed that blend films based on carboxymethyl cellulose (CMC) and GEL offered quick release of ASX (‘burst effect’), while the octenyl succinic anhydride starch (OSA)/GEL system ensured ASX release with a time lag [8]. Another study showed that films based on water-soluble soy polysaccharides (WSSPs) and GEL demonstrated a more proportional ASX release rate in comparison to the gum Arabic (GAR)/GEL system. As oxidation reactions occur mostly on food surfaces, the controlled release of ASX may be beneficial, as the antioxidant will persist on the food surface (where it is most needed) for an extended period of time [11]. Depending on the concentration, the active substances incorporated into biopolymer-based packaging materials may improve or worsen their functionality. Furthermore, the different packaging materials may ensure various degrees of stability of active ingredients. 

Therefore, this study aimed to assess the influence of carrier type (75/25 blends of CMC/GEL, GAR/GEL, OSA/GEL, and WSSP/GEL) and AST concentration (0, 0.25, 0.5, and 1% *w*/*w*) on the functional (light transmission, tensile properties, stability of colour, and stability of antioxidant activity) characteristics of the resulting edible active films. The data given here complement the previous publications [8,11], which mainly focused on the kinetics and mechanisms of AST release from the films. 

## 2. Materials and Methods

### 2.1. Materials 

The following food-grade polymers were used in this study: sodium CMC WALOCEL CRT 30 GA (viscosity: 25–35 cP (2% solution 25 °C), a substitution degree of 0.82–0.95; Dow Wolff Cellulosics, Bomlitz, Germany), GAR Agri-Spray Acacia R (Agrigum International, High Wycombe, UK), waxy maize starch sodium octenyl succinate Purity Gum^®^ 2000 (Ingredion, Hamburg, Germany), WSSP (Gushen Biological Technology Group Co., Dezhou, China), and pork GEL type A (pI ≈ 9, moisture content: 10.5%) with a Bloom strength of 240 (McCormick-Kamis S.A., Stefanowo, Poland). The AstaSana™ 5% CWS/S-TG formulation (containing astaxanthin, modified starch, corn starch, glucose syrup, sodium ascorbate, and DL-α-tocopherol) was supplied by DSM Nutritional Products (Venlo, The Netherlands). Glycerol (purity > 99%, used as a plasticizer) and 2,2′-azino-bis(3-ethylbenzothiazoline-6-sulfonic acid) diammonium salt (ABTS) were obtained from Sigma-Aldrich (Saint Louis, MO, USA).

### 2.2. Film Preparation

Films were prepared from aqueous biopolymer solutions containing a mixture of polysaccharides (3.75% *w*/*w*), GEL (1.25% *w*/*w*), glycerol (1% *w*/*w*), and increasing amounts of AST (0, 0.25, 0.5, and 1% *w*/*w*). Initially, powder blends of polysaccharides with GEL were weighed in ratios of 75/25, dissolved in water, and mixed with glycerol. The obtained film-forming solutions (FFSs, 150 g) were heated in closed 250 mL glass bottles placed in a water bath at 90 °C for 1 h with regular mixing by vigorous manual shaking. After removing from the bath, the FFSs were continuously shaken (200 rpm, Heidolph Unimax 1010, Schwabach, Germany) to cool them down to ~40 °C. Then, the AST was introduced and the FFSs were mixed (30,000 rpm, 2 min) using an MT-30K hand-held homogenizer (Hangzhou Miu Instruments Co., Ltd., Hangzhou, China). Afterward, the FFSs were degassed (with 115-mesh stainless-steel sieve) and placed on polycarbonate trays with the area of 144 and 4 cm^2^. A constant amount of total solids (0.0125 g/cm^2^) was cast and placed on the trays. The FFSs were dried at 25 ± 2 °C and 50 ± 5% relative humidity (RH) for 24 h. Then, the films were cut into samples and conditioned (25 °C, 50 RH, 48 h; MLR-350, Sanyo Electric Biomedical Co., Ltd., Oizumi-machi, Japan).

### 2.3. Microscopic Analysis

The microstructure of the films was observed using a light microscope (Leica5500B Microsystems GmbH, Wetzlar, Germany) with the use of a differential interference contrast (DIC) optical system and polarised light.

### 2.4. Moisture Content (MC)

The specimens (4 cm^2^) were dried in an oven at 105 °C for 24 h. The MC was calculated as the percentage of water removed from the system.

### 2.5. Mechanical Properties

Tensile strength (TS), elongation at break (E), and elastic modulus (EM) were determined using a TA-XT2i texture analyser equipped with a 50 kg load cell (Stable Micro Systems, Godalming, UK). Film samples (20 mm × 50 mm) were mounted on the analyser with an initial grip separation at 30 mm and stretched with a speed of 1 mm s^−1^. TS (MPa), E (%), and EM (MPa) were calculated using Equations (1)–(3), respectively:TS = F_max_/A(1)
where F_max_ is the maximum load for breaking the film (N) and A is the initial specimen cross-sectional area (thickness × width, mm^2^).
E = (ΔL/L) × 100(2)
where L is the initial gage length (mm) and ΔL is the difference in the length at the moment of fracture.
EM = (σ_2_ − σ_1_)/(ε_2_ − ε_1_)(3)
where ε_1_ is a strain of 0.2 (0.67%), ε_2_ is a strain of 0.4 (1,33%), σ1 (MPa) is the stress at ε_1_, and σ_2_ (MPa) is the stress at ε_2_.

### 2.6. Light Transmission (T)

The T (%) of films (10 mm × 50 mm) was measured in the wavelength range of 200–700 nm using a spectrophotometer (Lambda 40, Perkin-Elmer, Shelton, CT, USA). Five replicates of each film type were tested for the determination of T. 

### 2.7. Colour Stability 

Films (10 mm × 50 mm) were stored without light at 25 ± 1 °C and ~50% RH for 60 days in the test chamber (MLR-350, Sanyo Electric Biomedical Co., Ltd., Oizumi-machi Japan). The colour parameters (L*a*b*) of the films were measured with a colorimeter (NH310, 3nh, China) at three different time points (0, 30, and 60 days) using a white background (L* = 93.83, a* = 0.34, b* = −10.41). Moreover, the accelerated storage condition (at 60 °C, three time points: 0, 15, and 30 days) was employed. The analyses were performed in five replications.

Changes in the total colour differences (∆E) of the films during storage were calculated using Equation (4).
(4)$$\Delta E=\sqrt{{\left(\Delta L\right)}^{2}+{\left(\Delta a\right)}^{2}+{\left(\Delta b\right)}^{2}}$$

### 2.8. Impact of Storage on the Antioxidant Activity of the Films

The antiradical activity of the films (4 cm^2^) during storage (without light, 25 ± 1 °C, 50% RH; three time points: 0, 30, and 60 days) was determined according to a modified procedure proposed by Re et al. [12]. Fifty milligrams of film were dissolved in 1.5 mL of H_2_O (40 °C). An amount of 40 μL of the sample solution was added to 1.96 µL of the ABTS*^+^ solution and the absorbance was measured using the spectrophotometer (Lambda 40, Perkin-Elmer, Shelton, CT, USA) at 734 nm after 60 min. ABTS*^+^ scavenging was calculated according to Equation (5).
Scavenging (%) = [1 − (Abs/Abs_ABTS_)] × 100 (5)
where Abs is the absorbance of the sample and Abs_ABTS_ is the absorbance of the ABTS*^+^ solution (0.70 ± 0.05). The tests were performed in triplicate.

### 2.9. Statistical Analysis

Differences among the mean values of the data were tested for statistical significance at the *p* < 0.05 level using analysis of variance (STATISTICA 13.1, StatSoft Inc., Tulsa, OK, USA) and Fisher’s test.

## 3. Results and Discussion

### 3.1. Microstructure

Figure 1 shows micrographs of the control and 1% AST-supplemented films. The microstructures of the AST-free GAR- and OSA-based films were homogeneous and smooth (Figure 1A), which indicates a good compatibility between the polysaccharides and GEL. In turn, the CMC- and WSSP-based films exhibited rough topography. The inhomogeneity of the CMC/GEL film can be attributed to the immiscibility of its biopolymeric components manifested as phase-separated GEL-rich microspheres [8,10].

In turn, the grainy structure of the WSSP-based film reflects the presence of micro-sized polymer particles in the solution (Appendix A). In spite of nearly 100% solubility (Appendix A), the dissolved fraction of WSSP existed as microparticles (maximum diameter ≈ 4 μm). Moreover, it was found that the presence of GEL slightly decreased the solubility of WSSP (Appendix A), which could be the result of the association of WSSP with GEL [13], manifested by the formation of microscopic agglomerates (Appendix A). To support this observation, our previous study also showed that the blend WSSP/GEL film was more rough than the neat WSSP film [14].

The AST-containing films were intensively red (Figure 1B). Moreover, the active films were uneven (Figure 1B) as a result of the presence of starch granules in the AST formulation (as evidenced by the occurrence of Maltese crosses, Figure 1C).

### 3.2. MC

As can be seen from Table 1, the CMC- and OSA-containing films had a higher MC (~14%) than the GAR- and WSSP-based carriers (~11%), which is partially consistent with the result of a previous study [14]. Regardless of the carrier type, the increasing concentration of AST tended to reduce the MC value. Considering the fact that all films were obtained from the same amount of total solids (to achieve the same film thickness), this result cannot be explained by the increasing dry mass content in the films. Based on the results of a previous study [15], it can be speculated that AST affected the moisture retention in the carriers through the development of hydrogen bonds between the polymers and the AST constituents, which inhibited the formation of polymer-water hydrogen bonds and, consequently, resulted in limited MC. Among the AST-supplemented samples, the CMC-based films had the highest MC. This result may be attributed to the presence of numerous hydrophilic groups (hydroxyl and carboxylic) in the CMC structure. The excellent water binding and moisture sorption properties of CMC-based films have been proved previously [10,16].

### 3.3. Mechanical Properties

Regardless of the AST concentration, CMC yielded films with the best mechanical strength and stiffness (Table 1), which partially supports the evidence from previous observations [8]. Specifically, compared to the WSSP-, GAR-, and OSA-based samples, the TS of the CMC/GEL films was approx. 3, 3.5, and 13 times higher, respectively. This result can be easily explained by differences in the three-dimensional arrangements of the polysaccharides used for preparation of the films. It is well known that linear polymers, such as cellulose and its derivatives, form intra- and interchain H-bonds, which cause them to adhere strongly, resulting in a highly cohesive matrix [8,17]. In contrast, the branched polymers (in this study, GAR, waxy OSA, and WSSP) offer reduced packing efficiency [18]. Apart from that, the weak mechanical strength of GAR- and WSSP-based films can be attributed to the heterogeneous characteristics of these polysaccharides. GAR is a mixture of long and short chains of sugars (arabinogalactan, oligosaccharides, and polysaccharides) and glycoprotein [19], while WSSP is primarily a mixture of rhamnogalacturonans and arabinogalactans [20]. It has been suggested that, similar to polymer branching, the polymer heterogeneity can also give a material with reduced cohesiveness [14]. 

It is known that amorphous and crystalline polymers have irregularly and uniformly packed molecules, respectively. Therefore, compared to amorphous materials, crystalline materials are harder. As has been verified by wide X-ray diffraction analysis, GAR-, OSA-, and WSSP-containing films are amorphous [14], while CMC-containing films exhibit a semi-crystalline structure [21]. These structural differences provide an additional explanation for the tensile behaviour of the films (Table 1).

It should be mentioned that despite the phase-separated morphology of the CMC/GEL films [14], their mechanical resistance was better than other blend materials (Table 1). It can be ascribed to the fact that CMC as a predominant (continuous) matrix component ensured the strong polysaccharide–polysaccharide interactions, so the local presence of a dispersed phase (GEL) was of secondary importance.

The effect of AST on the mechanical properties was dependent on the type of carrier (Table 1). It was found that only in the case of the CMC/GEL film, AST reduced the TS value, suggesting that it interfered in the formation of a coherent matrix. In turn, AST did not affect the TS and EM of the OSA- and WSSP-based films (Table 1). These results indicate that a less organized network (created by the branched polymers) is not susceptible to mechanical changes caused by the presence of the active compound, while the more ordered network of CMC is weakened, possibly due to the interruption of hydrogen bonding between the polymer chains [8]. Regarding the GAR-based films, the low (0.25%) and moderate (0.5%) AST concentrations did not affect the TS values; however, the inclusion of AST at the highest level (1%) caused the mechanical strength of the carrier to increase (from 10.31 to 15.62 MPa). Moreover, an accompanying increase in EM was observed. It is difficult to explain this result, but it might be attributed to the binding/filling action of the bulky amounts of AST. Such an effect has been suggested previously to explain the decreased solubility of an AST-supplemented GAR-based film [11]. The possibility of enhancing mechanical strength may be related to the fact that GAR forms very frangible films that do not have sufficient mechanical resistance to be applied [14,22]. The mixing of GAR with GEL (at a ratio of 3:1) significantly improves the cohesiveness of films; however, the material durability is still relatively low [14]. In accordance with the present results, previous studies on ascorbyl-palmitate-supplemented films have also demonstrated that it is possible to improve the mechanical integrity of the GAR/GEL matrix [9]. 

Consistent with the previous studies [9,14], OSA yielded the most stretchable films with the lowest resistance to deformation (E = 108.08–139.21%, EM = 16.16–22.98 MPa; Table 1). This result may be ascribed to the strong plasticizing action of octenyl succinic groups on amylopectin chains. Usually, films obtained from glycerol-plasticized unmodified waxy maize starch exhibit several times lower E [23] and tens of times greater stiffness [24].

### 3.4. T

Considering the transparency to visible light, the control samples were ranked in the following order: OSA/GEL > GAR/GEL >WSSP/GEL > CMC/GEL films (Figure 2). These differences reflect primarily the microstructural features of the films. As the OSA- and GAR-based films were fairly homogenous (Figure 1), they offered the highest transparency. In turn, the roughness of the CMC- and WSSP-based films was responsible for their partial opaqueness. The CMC-based film had the lowest T_λ400–700_ (57.96–78.90%, Figure 2), probably due to the presence of phase-separated microspheres [8,10], which scattered, reflected, or absorbed light. As a result of the inhomogeneous microstructure, the CMC- and WSSP-based films exhibited the best UV-C barrier properties. Namely, at λ = 280, the T values of the WSSP-, CMC-, GAR-, and OSA-based films were approx. 8%, 11%, 32%, and 39%, respectively (Figure 2). The better UV blocking properties of the WSSP/GEL film compared to the CMC/GEL film may be ascribed to the fact that WSSP contains a small fraction (~5–14%) of protein [25,26]. Regardless of the polysaccharide type, all blend films completely blocked UV light below λ ≈ 240 nm, which can mainly be related to the excellent UV protection property of GEL [27,28].

Generally, as the AST concentration increased, the T value successively decreased (Figure 2). This result may be partially explained by the presence of the increasing content of starch granules (from the AST formulation (Figure 1C)) that reflected/scattered a significant amount of light. In the spectra of the 0.25 and 0.5% AST-supplemented films, the observed transmittance band centred at λ ≈ 355 nm (Figure 2) was a consequence of light absorption in the region λ ≈ 450–500 nm (data not shown). This finding is not surprising, considering the fact that the absorption maxima of ASX are situated between 409 and 476 nm [29]. It was found that the 1% AST-supplemented films effectively blocked the light <λ ≈ 580 nm. This outcome can be easily explained by the intense dark colour of these films (Appendix A). 

### 3.5. Colour Stability at Room Temperature

To evaluate the colour stability of the obtained films, the storage experiment (25 °C for 60 days, without light) was conducted. Among the AST-free films, the WSSP-based carrier was the most prone to colour change (∆E_60days_ = 3.70). It is difficult to explain this finding; however, it should be mentioned that the WSSP/GEL film was initially the most yellow and dark (Appendix A). Therefore, it is possible that the colour change was caused by the darkening of the pigments present in the polysaccharide fraction. Among the control samples, the best colour stability was observed for the CMC- and GAR-based films. 

The AST-supplemented samples were darker, redder, and more yellowish than the controls (Appendix A). The gradual increase in the AST concentration resulted in a gradual reduction in the lightness of the films (Appendix A). Surprisingly, the highest values of the a* (redness) and b* (yellowness) parameters were observed for the 0.25% AST-supplemented films (Appendix A), which agrees with our earlier observations [11]. As mentioned previously, these films were not overloaded with dye; thus, they exhibited the purest bright red colour. The CMC/GEL films incorporated with the higher level of AST (0.5–1%) exhibited the highest redness among all samples.

Regardless of the AST concentration, the AST-supplemented CMC- and GAR-based films exhibited the lowest colour change after 60 days of storage (∆E < 2, Table 2). Such a small colour difference could be noticed only by an experienced observer [30]. Regarding the OSA/GEL films, in some cases, ∆E exceeded 2, which also means that an inexperienced observer can notice the difference. A clear colour change was observed in the WSSP/GEL films (in some cases, ∆E exceeded 3.5, Table 2). The high ∆E values of the WSSP-based films were mainly caused by the decrease in the L* and a* parameters (Appendix A). This result suggests that ASX was the least stable in the WSSP/GEL carrier.

### 3.6. Accelerated Colour Stability Test

As relatively small changes in the colour of the samples (∆E < 5, Table 2) were detected after storage at 25 °C, the accelerated colour stability test (at 60 °C) was employed in order to gain better insight in colour retention in the particular carriers. As expected, the obtained ∆E values were much higher (Table 2). Similar to the room storage test, the control CMC-based film was the most stable in terms of colour changes (∆E < 3). This was mainly related to the fact that this film was the least darkened and yellowed (Appendix A) during storage.

In the case of the other control samples, the colour changes were more noticeable (∆E_15days_ = 4.78–7.72) and deepened over time (∆E_30days_ = 9.06–10.59, Table 2). The weak colour changes observed in the control CMC/GEL film may be explained by the immiscibility of its constituents (Figure 1A). It can be assumed that, in the phase-separated polysaccharide/protein blend system, the intensity of the Maillard reaction was lower than in the other obtained carriers. The possibility of occurrence of the Maillard reaction between GAR and proteins, as well as WSSP and proteins, has been previously proved [31,32]. The available literature does not provide data on the possibility of participation of OSA in Maillard reactions; however, the results of this research indicate such an interaction.

As can be seen from the data in Table 2, with the increasing concentration of AST, the susceptibility to colour change tended to decrease. This indirectly indicates that the higher the ASX amount in the films, the better its stability. Moreover, it may be concluded that, when AST was incorporated at the lowest level, the colour changes were easier to detect than in the films overloaded with the pigment. Among the 0.25% AST-supplemented films, the weakest colour stability (∆E_30days_ = 19.52, the highest L* and b* values) was observed for the WSSP-based film. In turn, at the highest AST addition level (1%), the CMC/GEL film tended to be the best carrier in terms of colour stability (Table 2). These results suggest that the type of the polysaccharide-based carrier affects the stability of ASX during storage.

### 3.7. Impact of Storage on the Antioxidant Activity of the Films

As can be seen in Figure 3, the control polysaccharide/GEL films (after complete dissolution in water at 40 °C) showed a relatively high ABTS^+^* scavenging activity (~42–46% at 60 min of incubation). Considering the fact that purified polysaccharide fractions have slight antioxidant potential [33], the observed result may be associated with the presence of the GEL fraction in the films. As reported in previous studies, the high glycine and proline contents in GEL are responsible for its good neutralisation of free radicals [34,35]. Regardless of the carrier type, the storage time (25 °C, 60 days) did not affect the antioxidant properties of the control samples. Generally, the increase in the AST concentration resulted in the increased antioxidant potential of the films. After the incorporation of 1% AST, the antiradical activity of the films increased by 1.5 times (~25 percentage points) compared to the controls. This result is not very impressive. However, it should be mentioned that the AST used in this study contains approx. 5% of ASX (DSM, 2014), so the real concentration of ASX in the FFSs at the highest AST addition level was only 0.05%.

Regardless of the AST concentration, the WSSP-containing films had the weakest antiradical activity. It is difficult to explain this result, but it might be related to the faster nonenzymatic browning rate in this system, as evidenced by the intensive colour change (Table 2), especially darkening of the samples (decrease in L* values) during storage (Appendix A). The possibility of occurrence of the Maillard reaction between WSSP and proteins (at 55–65 °C, for 36–96 h) has been previously proven [32]. It can be speculated that the blocking of the amino groups of GEL by the formation of polysaccharide/GEL conjugates resulted in weaker antioxidant activity of the obtained materials.

Generally, it was found that the films with the lowest and highest AST contents were the most and least stable in terms of antioxidant potential, respectively.

## 4. Conclusions

The CMC/GEL films exhibited at least twice as high TS as the GAR/GEL and WSSP/GEL films, and ten times higher TS than the OSA/GEL films. This outcome may be ascribed to the linear structure of CMC, which facilitated the formation of a more cohesive matrix, compared to the mixture of GEL with the branched polysaccharides. The more ordered network of the CMC-based film, however, was susceptible to mechanical weakening in the presence of AST (the TS value was reduced by ~14%). In turn, the bulky amounts of AST (1%) acted as a binder/filler that improved the TS of the GAR-based film. The increasing concentration of AST resulted in a decline in the transparency of the films, which may be associated with the presence of starch granules (carrying component) in the AST formulation.

Generally, the increase in the AST concentration resulted in darker films. As a result, the films with the lowest AST content (0.25%) were the most red. As expected, the films with the highest AST concentration (1%) exhibited the best antiradical capacity. The 0.25% AST-supplemented films were the most stable in terms of the antioxidant potential, while the 1% AST-supplemented films were the most prone to deterioration of the antiradical activity during storage.

Despite their high transparency, the excessive plasticity (mechanical weakness) of the OSA-based films could make their application difficult. Considering their weakest colour stability and the weakest antiradical capacity, the WSSP-based film is the least desirable carrier for AST. In turn, the CMC/GEL film exhibited the most stable colour during storage (the lowest rate of darkening and yellowing) at both room and elevated temperatures. This result may be associated with the fact that GEL and CMC are immiscible polymers, as verified by microscopic observations, which probably limited the possibility of occurrence of the Maillard reaction. However, a serious drawback of the CMC/GEL AST-supplemented films was their low light transparency (due to the surface roughness). The GAR/GEL system was devoid of this disadvantage and was also quite stable during storage; therefore, it can be taken into account for some food packaging purposes that do not require high mechanical strength (e.g., coatings).

## Figures and Tables

**Figure 1 polymers-14-04001-f001:**
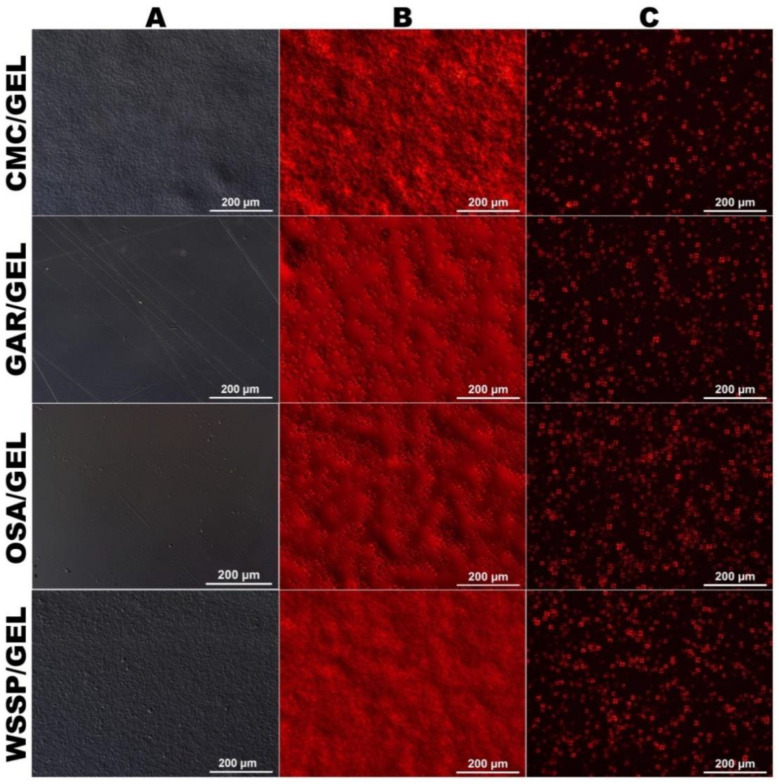
Microtopographies of AstaSana astaxanthin (AST)-free (**A**) and 1% AST-supplemented films (**B**,**C**) obtained from the 75/25 blends of polysaccharides (carboxymethyl cellulose (CMC), gum Arabic (GAR), octenyl succinic anhydride starch (OSA), and water-soluble soy polysaccharides (WSSP)) and gelatin (GEL). (**A**,**B**)—unpolarized light; (**C**)—polarized light.

**Figure 2 polymers-14-04001-f002:**
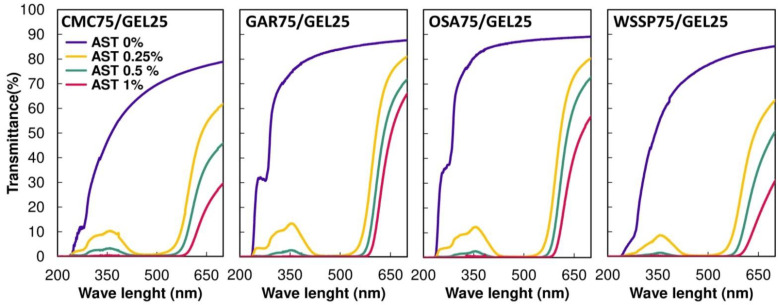
Effect of the increasing concentration of AstaSana astaxanthin (AST) on the light transmission of the films obtained from the 75/25 blends of polysaccharides (carboxymethyl cellulose (CMC), gum Arabic (GAR), octenyl succinic anhydride starch (OSA), and water-soluble soy polysaccharides (WSSP)) and gelatin (GEL).

**Figure 3 polymers-14-04001-f003:**
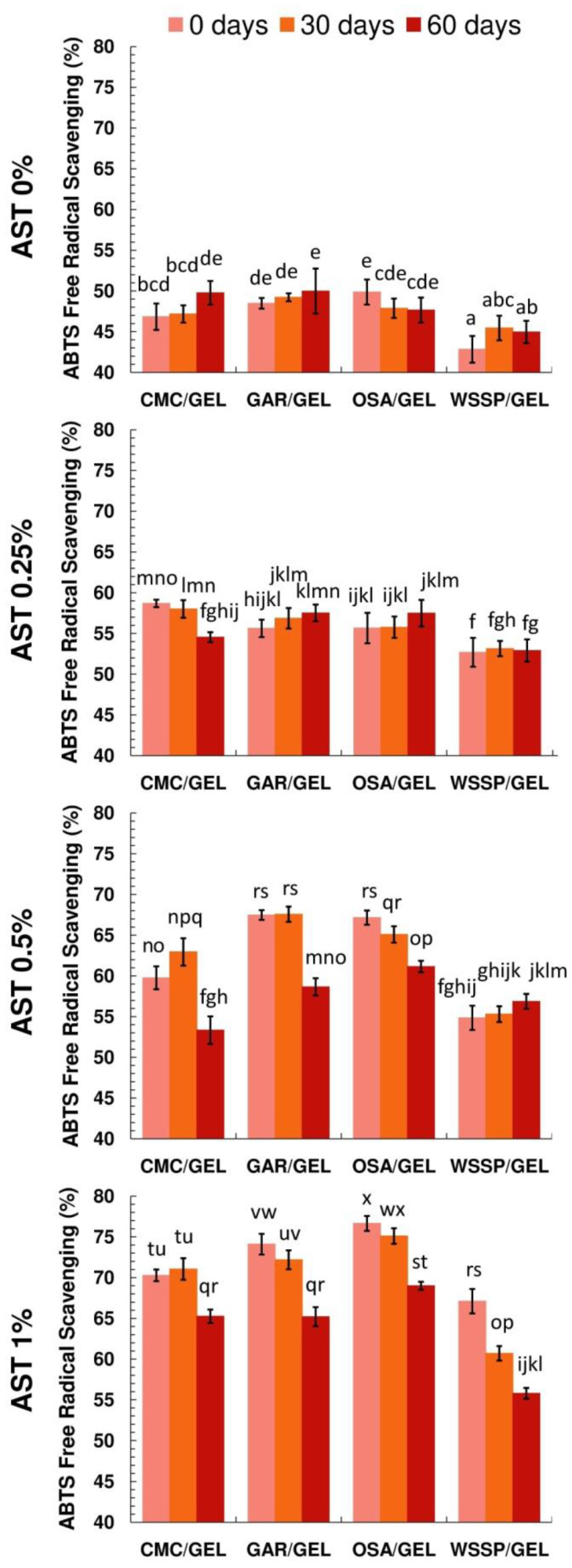
Effect of storage (25 °C, 50% relative humidity) on the antiradical activity of AstaSana astaxanthin (AST)-free (A) and AST-supplemented films obtained from the 75/25 blends of polysaccharides (carboxymethyl cellulose (CMC), gum Arabic (GAR), octenyl succinic anhydride starch (OSA), water-soluble soy polysaccharides (WSSP)) and gelatin (GEL). a–x Values with the different superscript letters within one column are significantly different (*p* > 0.05).

**Table 1 polymers-14-04001-t001:** The effect of Astasana (AST) concentration on the moisture content (MC), tensile strength (TS), elongation at break (E), and elastic modulus (EM) of 75/25 blend films obtained from polysaccharides: carboxymethyl cellulose (CMC), gum Arabic (GAR), octenyl-succinic-anhydride-modified starch (OSA), water-soluble soy polysaccharides (WSSP), and gelatin (GEL).

Films	AST (%)	MC (%)	TS (MPa)	E (%)	EM (MPa)
CMC/GEL	0	14.28 ± 0.21 ^i^	43.04 ± 3.81 ^h^	22.59 ± 6.53 ^b^	1242.68 ± 179.48 ^fg^
	0.25	13.62 ± 0.31 ^h^	38.36 ± 1.61 ^g^	20.51 ± 7.06 ^ab^	1302.23 ± 112.87 ^g^
	0.5	12.78 ± 0.24 ^g^	37.51 ± 2.87 ^g^	25.20 ± 4.93 ^bc^	1171.33 ± 135.84 ^e^
	1	12.03 ± 0.24 ^f^	36.88 ± 2.78 ^g^	12.57 ± 2.94 ^ab^	1216.89 ± 146.18 ^ef^
GAR/GEL	0	11.58 ± 0.46 ^ef^	10.31 ± 2.06 ^bcd^	15.31 ± 4.46 ^ab^	314.90 ± 96.39 ^b^
	0.25	11.18 ± 0.43 ^de^	9.09 ± 2.05 ^b^	10.21 ± 4.07 ^a^	231.01 ± 98.31 ^b^
	0.5	10.95 ± 0.24 ^d^	10.28 ± 3.61 ^bcd^	8.89 ± 4.31 ^a^	423.41 ± 114.67 ^c^
	1	10.14 ± 0.21 ^bc^	15.62 ± 1.57 ^f^	14.16 ± 4.13 ^ab^	673.45 ± 80.14 ^d^
OSA/GEL	0	14.62 ± 0.21 ^i^	3.26 ± 0.53 ^a^	139.21 ± 25.21 ^g^	16.48 ± 5.77 ^a^
	0.25	12.66 ± 0.64 ^g^	3.07 ± 0.36 ^a^	132.08 ± 14.49 ^fg^	16.16 ± 4.31 ^a^
	0.5	11.81 ± 0.66 ^f^	3.17 ± 0.60 ^a^	124.53 ± 22.43 ^f^	19.55 ± 4.87 ^a^
	1	10.69 ± 0.48 ^cd^	2.69 ± 0.26 ^a^	108.08 ± 14.75 ^e^	22.98 ± 4.08 ^a^
WSSP/GEL	0	11.01 ± 0.48 ^de^	13.05 ± 1.47 ^e^	39.36 ± 8.07 ^d^	497.80 ± 125.45 ^c^
	0.25	10.01 ± 0.50 ^b^	11.63 ± 2.26 ^cde^	34.56 ± 11.45 ^cd^	495.57 ± 148.84 ^c^
	0.5	9.30 ± 0.31 ^a^	13.03 ± 2.03 ^e^	39.33 ± 9.45 ^d^	421.01 ± 86.66 ^c^
	1	9.33 ± 0.53 ^a^	12.12 ± 1.63 ^de^	32.88 ± 8.92 ^cd^	412.73 ± 43.07 ^c^

^a^^–i^ Values with the different superscript letters within one column are significantly different (*p* > 0.05).

**Table 2 polymers-14-04001-t002:** The total colour differences during storage (ΔE) of 75/25 blend films obtained from polysaccharides: carboxymethyl cellulose (CMC), gum Arabic (GAR), octenyl succinic anhydride-modified starch (OSA), water-soluble soy polysaccharides (WSSP), and gelatin (GEL) incorporated increasing Astasana (AST) concentration.

Films	AST (%)	25 °C		60 °C	
		30 Days	60 Days	30 Days	60 Days
CMC/GEL	0	0.88 ± 0.12 ^abc^	0.98 ± 0.21 ^abcde^	2.24 ± 0.24 ^a^	2.73 ± 0.38 ^a^
	0.25	0.91 ± 0.11 ^abcd^	1.24 ± 0.20 ^cdefg^	11.10 ± 0.34 ^jkl^	13.40 ± 0.56 ^op^
	0.5	1.45 ± 0.32 ^fghi^	1.56 ± 0.23 ^ghij^	5.26 ± 0.37 ^bc^	5.38 ± 0.49 ^bc^
	1	1.11 ± 0.30 ^cdefg^	0.94 ± 0.16 ^abcde^	6.23 ± 0.31 ^cd^	6.79 ± 1.08 ^de^
GAR/GEL	0	0.60 ± 0.14 ^a^	1.09 ± 0.18 ^bcdef^	7.72 ± 0.40 ^ef^	10.59 ± 0.75 ^ijk^
	0.25	0.64 ± 0.17 ^ab^	1.66 ± 0.19 ^hij^	11.89 ± 0.16 ^lm^	13.67 ± 0.96 ^opq^
	0.5	0.58 ± 0.16 ^a^	1.37 ± 0.20 ^bcd^	11.25 ± 0.23 ^kl^	11.99 ± 0.48 ^lm^
	1	1.16 ± 0.23 ^cda^	1.13 ± 0.18 ^ab^	7.95 ± 0.71 ^efg^	8.24 ± 0.59 ^fg^
OSA/GEL	0	1.35 ± 0.07 ^defgh^	1.88 ± 0.12 ^ijk^	4.78 ± 0.59 ^b^	9.06 ± 0.40 ^gh^
	0.25	1.46 ± 0.13 ^fghij^	2.31 ± 0.23 ^kl^	12.76 ± 0.27 ^mno^	15.50 ± 0.71 ^r^
	0.5	1.45 ± 0.47 ^fghi^	2.54 ± 0.16 ^l^	11.10 ± 0.31 ^jkl^	14.31 ± 0.57 ^pqr^
	1	1.25 ± 0.21 ^cdefg^	1.91 ± 0.25 ^jk^	12.10 ± 0.39 ^lmn^	14.34 ± 1.11 ^pqr^
WSSP/GEL	0	2.57 ± 0.17 ^l^	3.70 ± 0.14 ^m^	7.23 ± 0.63 ^def^	9.73 ± 1.47 ^hi^
	0.25	2.70 ± 0.34 ^l^	1.20 ± 0.20 ^cdefg^	17.20 ± 0.36 ^s^	19.52 ± 1.02 ^t^
	0.5	3.67 ± 0.31 ^m^	2.54 ± 0.16 ^g^	13.30 ± 0.37 ^nop^	14.65 ± 0.91 ^qr^
	1	4.41 ± 0.34 ^n^	4.02 ± 0.20 ^mn^	9.74 ± 0.37 ^hi^	10.03 ± 1.63 ^hij^

^a–t^ Values with the different superscript letters within one column are significantly different (*p* > 0.05).

## Data Availability

The data present in this study are available on request from the first author (Katarzyna Łupina).

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
