# Peer review of "Functional Properties and Storage Stability of Astaxanthin-Loaded Polysaccharide/Gelatin Blend Films—A Comparative Study"

_polymers, 2022, doi:10.3390/polym14194001_

Round 1

Reviewer 1 Report

Authors must improve their manuscript before it can be considered for publication in the now prestigious Polymers. In this regard, authors are encouraged to address the following issues:

1. The objective of the work must clearly be declared in an independent paragraph.

2. What is the role of using glycerol? Is this a food-grade chemical?

3. Authors claim that there is limited solubility of WSSP in GEL based on the grainy morphology of films. Please elaborate on actual solubility data and state the concentration solubility limits. If this is the case, what's the point of preparing and presenting an already-known immiscible blend?

4. How do the authors explain the best mechanical properties of CMC-GEL when compared to other formulations if they already reported limited blend miscibility? This does not make any sense. Please change the discussion to a more scientifically-sound argument.

5. The less transparent film is CMC-GEL according to the surface topography but as opposed to their mechanical properties. Please discuss these issues accordingly.

6. FTIR and XRD must be presented to shed light on the nature of molecular interactions and semi-crystalline-to-amorphous transitions (if any) from pristine polysaccharides to blending.

7. Based on the above arguments, the conclusions are not well supported by the experimental evidence presented.

In general, the paper is interesting but it lacks a fundamental understanding in correlating structural to performance features. The paper must be improved before it can be considered for publication in Polymers.

Author Response

Answers to Reviewer 1

Authors must improve their manuscript before it can be considered for publication in the now prestigious Polymers. In this regard, authors are encouraged to address the following issues:

  1. The objective of the work must clearly be declared in an independent paragraph.

As suggested by the Reviewer, the purpose of the study has been clarified and declared in an independent paragraph. Please see L73-78.

  1. What is the role of using glycerol? Is this a food-grade chemical?

In the present study, glycerol (purity > 99% = molecular biology grade) was used as a plasticizer. It has been mentioned in the revised manuscript (Please see L89-90). Without glycerol it is often not possible to obtain coherent (non-cracked) and flexible biopolymeric films. The EFSA approved glycerol as a safe food additive (E422). https://www.efsa.europa.eu/en/efsajournal/pub/4720

  1. Authors claim that there is limited solubility of WSSP in GEL based on the grainy morphology of films. Please elaborate on actual solubility data and state the concentration solubility limits. If this is the case, what's the point of preparing and presenting an already-known immiscible blend?

The solubility of the WSSP has been determined in range of 1-10% w/w, without and with presence of gelatin. Please see Figure S1. Moreover, microscopic analyses of the supernatants has been presented. Please see Figure S2. The explanation of the morphology of WSSP-based films has been improved. Please see L173-180.

  1. How do the authors explain the best mechanical properties of CMC-GEL when compared to other formulations if they already reported limited blend miscibility? This does not make any sense. Please change the discussion to a more scientifically-sound argument.

As has been already mentioned, the best tensile strength and stiffens of the CMC75/GEL25 films could be attributed to the fact that the linear character of CMC offers better cohesiveness than the branched GAR, OSA, and WSSP. Please see L212-213. Moreover, it could be assumed that CMC as a predominant (continuous) matrix component ensured the strong polysaccharide-polysaccharide interactions, so the local presence of dispersed phase (GEL) was of secondary importance. Please see L220-230.

  1. The less transparent film is CMC-GEL according to the surface topography but as opposed to their mechanical properties. Please discuss these issues accordingly.

As mentioned before the incompatibility of the filmogenic ingredients does not prejudge the mechanical weakness of the blend material. It has been mentioned in the revised manuscript (Please see L226-230). We agree with the Reviewer that the microstructural properties could affect both optical and mechanical properties; however, generally there is no correlation between the optical and mechanical properties.

  1. FTIR and XRD must be presented to shed light on the nature of molecular interactions and semi-crystalline-to-amorphous transitions (if any) from pristine polysaccharides to blending.

The Reviewer is right, however, the FTIR spectra of the obtained films have been presented and discussed in the previous two works:

  • Gum Arabic/Gelatin and Water-Soluble Soy Polysaccharides/Gelatin Blend Films as Carriers of Astaxanthin—A Comparative Study of the Kinetics of Release and Antioxidant Properties (https://www.mdpi.com/2073-4360/13/7/1062)
  • Controlled release of water-soluble astaxanthin from carboxymethyl cellulose/gelatin and octenyl succinic anhydride starch/gelatin blend films (https://www.sciencedirect.com/science/article/pii/S0268005X21005956)

Unfortunately, at the moment there are long waiting times for the WAXD analysis. We hope that the Reviewer will agree to the publication of manuscript without XRD results. It should be noted, however, that our previous work:

  • Edible films made from blends of gelatin and polysaccharide-based emulsifiers - A comparative study (https://www.sciencedirect.com/science/article/abs/pii/S0268005X19304965)

has presented the WAXD results of the 75/25 blend CMC/gelatin film.

Moreover, the diffractograms of the 50/50 blend GAR/gelatin, OSA/gelatin, and WSSP/gelatin films, as well as the single-polymer films have been previously presented and discussed.

  • Edible films based on gelatin, carboxymethyl cellulose, and their blends as carriers of potassium salts of iso-α-acids: Structural, physicochemical and antioxidant properties (https://www.sciencedirect.com/science/article/abs/pii/S0268005X20329489)

Therefore, the diffractograms of the control 75/25 polysaccharide/gel blend films are predictable. In the revised manuscript, the tensile behavior of the films has been additionally explained in terms of their structure previously verified by WAXD. Please see L220-225.

  1. Based on the above arguments, the conclusions are not well supported by the experimental evidence presented.

In the revised manuscript, the tensile behavior of the films has been additionally explained in terms of their structure previously verified by WAXD. Please see L220-225.

In general, the paper is interesting but it lacks a fundamental understanding in correlating structural to performance features. The paper must be improved before it can be considered for publication in Polymers.

The manuscript has been improved.

Reviewer 2 Report

The authors undertook the comparative study of the functional properties and storage stability of astaxan- 2 thin-loaded polysaccharide/gelatin blend films. I did not the novelty in the manuscript. 

1. Authors need to revised the abstract. 

2. SEM images should be incorporate. 

3. What about the porosity and tensile strength? Authors need to incorporate. 

4. A comparative table should be incorporated. 

5. Why authors choose astaxan- 2 thin-loaded polysaccharide and gelatin? Please explain in details. 

Author Response

Answers to Reviewer 2

The authors undertook the comparative study of the functional properties and storage stability of astaxanthin-loaded polysaccharide/gelatin blend films. I did not the novelty in the manuscript.

In our opinion the novelty of this work is determination of color stability and antioxidant stability of the astaxanthin-loaded into different biopolymeric films. Only comparative studies can help in selection of the best material for a given application

  1. Authors need to revised the abstract.

The abstract has been improved. the words limit has been reached.

  1. SEM images should be incorporate.

The reviewer is right, however, the SEM images of the films have been already presented in previous work “Mathematical Modeling of Water-Soluble Astaxanthin Release from Binary Polysaccharide/Gelatin Blend Matrices” https://www.mdpi.com/2504-5377/5/3/41

Therefore, in the present study we presented the differential interfered contrast images of the films. In our opinion these images provide deeper insight into the film’s microstructures that the SEM images.

  1. What about the porosity and tensile strength? Authors need to incorporate.

We thank you the Reviewer for indicating the possibility of explanation of the  mechanical strength of the films in terms of their porosity. Unfortunately we do not have access to a gas pycnometer. So, we can not comment of the porosity of the obtained materials. We hope that Reviewer will accept it. Perhaps we should be interested in buying a gas pycnometer in future.

  1. A comparative table should be incorporated.

We agree with the Reviewer that it is possible to present the data in one comparative table (8 columns). However, it could strongly limit the readability and clarity of the table, and could cause editorial problems. Therefore, we have decided to maintain two smaller tables with the data. We hope that Reviewer will accept it. 

5.Why authors choose astaxanthin-loaded polysaccharide and gelatin? Please explain in details.

The additional explanation has been added. Please see L51-56

Reviewer 3 Report

This study titled “Functional properties and storage stability of astaxanthin-loaded polysaccharide/gelatin blend films – a comparative 3 study” describes the preparation and characterization of polysaccharide/gelatin blend films prepared with different astaxanthin concentrations. The paper is very interesting, well written and well organized, and represents some advancement over the actual state-of-the-art. However, some revisions are required before it could be considered for publication, as follows:

- Abstract: Increasing concentrations (0, 0.25, 0.5, 1% .....what units?

- Abstract: The authors should add better experimental results obtained in this study. I recommend also briefly report the novelty of the present study.

- Introduction/last paragraph: How is this system different to previous reports [9,10] to merit publication? Please, report.

- Lines 62/63: Again, 0, 0.25, 0.5, 1% .....what units? I recommend a carefull revision of units in all manuscript.

- Lines 70/79: Some properties of CMC and gelatin such as molecular mass, viscosity and other relevant parameters should be reported. It is more intereting to the readers.

- Lines 80/91: The authors should report in detail the experimental procedure to prepare films, including final polymer solution volume, tank dimensions, mechanical or magnetic stirring and intensity, impeller used in its configuration.

Author Response

Answers to Reviewer 3

This study titled “Functional properties and storage stability of astaxanthin-loaded polysaccharide/gelatin blend films – a comparative study” describes the preparation and characterization of polysaccharide/gelatin blend films prepared with different astaxanthin concentrations. The paper is very interesting, well written and well organized, and represents some advancement over the actual state-of-the-art. However, some revisions are required before it could be considered for publication, as follows:

Abstract: Increasing concentrations (0, 0.25, 0.5, 1% .....what units?

The units used were % (w/w). It has been included in the revised manuscript.

Abstract: The authors should add better experimental results obtained in this study. I recommend also briefly report the novelty of the present study.

The abstract has been improved up to the words limit.

Introduction/last paragraph: How is this system different to previous reports [9,10] to merit publication? Please, report.

It has been mentioned in the revised manuscript. Please see L76-78.

Lines 62/63: Again, 0, 0.25, 0.5, 1% .....what units? I recommend a carefull revision of units in all manuscript.

The units are % w/w (g/100g). The units have been improved in the revised manuscript.

Lines 70/79: Some properties of CMC and gelatin such as molecular mass, viscosity and other relevant parameters should be reported. It is more intereting to the readers.

As requested by the Reviewer we have added the viscosity of the CMC and more details on gelatin. Please see L82-90. Unfortunately, molecular mass of the CMC and gelatin used is unknown. According to literature, gelatin has molecular weight ranging from 15 to 400 kDa, depending on manufacturing process and conditions used. Unfractionated gelatin is a mixture of polypeptide/polymer chains with different molecular weights including α-chains (~100 kDa), β-chains (~200 kDa), γ-chains (~300 kDa) and their  fragments.

Lines 80/91: The authors should report in detail the experimental procedure to prepare films, including final polymer solution volume, tank dimensions, mechanical or magnetic stirring and intensity, impeller used in its configuration.

The more details has been provided. Please see L93-103. The final concentration of the ingredients in the FFSs has been clarified. 150 g of FFSs was obtained at one time. It has been already mentioned.

Round 2

Reviewer 1 Report

Authors addressed all comments; the paper can be accepted for publication in Polymers

Reviewer 2 Report

Accept